# Evidence for ion migration in hybrid perovskite solar cells with minimal hysteresis

Philip Calado[1,2,*], Andrew M. Telford[1,*], Daniel Bryant[3,4], Xiaoe Li[3], Jenny Nelson[1,2,3], Brian C. O'Regan[5] & Piers R.F. Barnes[1,2]

Ion migration has been proposed as a possible cause of photovoltaic current–voltage hysteresis in hybrid perovskite solar cells. A major objection to this hypothesis is that hysteresis can be reduced by changing the interfacial contact materials; however, this is unlikely to significantly influence the behaviour of mobile ionic charge within the perovskite phase. Here, we show that the primary effects of ion migration can be observed regardless of whether the contacts were changed to give devices with or without significant hysteresis. Transient optoelectronic measurements combined with device simulations indicate that electric-field screening, consistent with ion migration, is similar in both high and low hysteresis $CH_3NH_3PbI_3$ cells. Simulation of the photovoltage and photocurrent transients shows that hysteresis requires the combination of both mobile ionic charge and recombination near the perovskite-contact interfaces. Passivating contact recombination results in higher photogenerated charge concentrations at forward bias which screen the ionic charge, reducing hysteresis.

[1] Department of Physics, Imperial College London, London SW7 2AZ, UK. [2] Centre for Plastic Electronics, Imperial College London, London SW7 2AZ, UK. [3] Department of Chemistry, Imperial College London, London SW7 2AZ, UK. [4] SPECIFIC, Swansea University, Swansea SA12 7AX, UK. [5] Sunlight Scientific, 1190 Oxford Street, Berkeley, California 94707, USA. * These authors contributed equally to this work. Correspondence and requests for materials should be addressed to B.C.O.R. (email: bor@borski.demon.co.uk) or to P.R.F.B. (email: piers.barnes@imperial.ac.uk).

L ead-halide perovskite solar cells have recently emerged as a promising new solution-processible photovoltaic technology. Despite rapid developments in efficiency figures and fabrication processes, the performance of many of these cells remains strongly dependent on their prior optical and electronic conditioning[1–3]. This was first referred to as an anomalous hysteresis in the characteristic current–voltage ($J$–$V$) scan in $CH_3NH_3PbI_3$ devices (known herein as simply hysteresis)[2]. Subsequently, the same process has been measured as a slow change in device photocurrent, photoluminescence intensity and open circuit voltage ($V_{oc}$) occurring on timescales up to hundreds of seconds[3–9]. Furthermore, its magnitude tends to increase with ageing/degradation[3,10–13]. Understanding this mechanism and its effect on photovoltaic performance is critical for directing future perovskite solar cell research to either resolve the issue or exploit it[7,14,15].

The leading model to explain hysteresis is that time-varying quantities of charge accumulated at the $CH_3NH_3PbI_3$ interfaces reduces or entirely screens the internal electric field, resulting in loss of photocurrent[1,6,7,16–20]. Migration of ionic defects in the perovskite phase, ferroelectric polarization or trapping of electrons at the interfaces have all been suggested as mechanisms for the accumulation of this charge[2]. However, it is not clear that ferroelectric polarization can persist in these materials[21,22]. The rate of any ferroelectric polarization/depolarization or the rate of trapping/detrapping of electrons at interfacial states would likely be too fast[17,21,23] relative to the slow timescales related to hysteresis phenomena (1–100 s) to be solely responsible[1–3,5,6]. There is strong direct and indirect evidence that slow drift and diffusion of ionic defects at room temperature is the dominant mechanism underlying hysteresis in $CH_3NH_3PbI_3$ solar cells[6,7,17,24–31]. However, a significant objection to this hypothesis is that the degree of hysteresis is highly dependent on the interface properties and choice of contact materials, which appear to control the interfacial trap density[2,19,32–40]. For example, when a ZnO cathode top layer is replaced with phenyl-C61-butyric acid methyl ester (PCBM) in a $CH_3NH_3PbI_3$ device with an otherwise identical architecture, hysteresis at room temperature is significantly reduced (see Fig. 1c and Supplementary Fig. 1). To a first approximation, ionic defect concentration and mobility in the bulk of the perovskite phase are not expected to be strongly influenced by the contacts, although the possibility that PCBM blocks ion migration at grain boundaries has been proposed[30]. Superficially, these observations appear to undermine the viability of the ion transport model for explaining hysteresis because the effect seems to be controlled by the contact material. However, recent simulations suggest that $J$–$V$ hysteresis could only be reproduced if both ion migration and recombination via interfacial traps were present in devices[16].

Here we present transient optoelectronic measurements that probe the direction of the internal electric field in operating devices. The measurements directly indicate that ionic migration appears in devices both with and without hysteresis. Our simulations reproduce the transient device behaviour over all relevant timescales ($10^{-8}$–$10^2$ s). The results show that hysteresis is only observed in cases where high rates of recombination exist in the perovskite/contact interfacial regions of devices. During the forward $J$–$V$ scan a reverse electric field in the bulk perovskite layer drives electrons and holes away from their respective transport layers and high concentrations of minority carriers build up at these interfaces. Where these interfaces act as recombination regions, the charge collection efficiency of the device is adversely affected. Instead, if recombination at the interfaces is reduced, then the build-up of photogenerated charge carriers contributes to efficient collection of diffusive currents

during the forward scan. Low hysteresis is thus primarily an artefact of low interfacial recombination and resultant high photogenerated carrier populations at forward bias, despite the presence of ion migration. Our evidence experimentally confirms the prediction of van Reenen et al.[16] and indicates that the stability of photocurrents and photovoltages in a device can be controlled by changing the interfacial recombination properties, without necessarily requiring a change in ion concentration or mobility within the perovskite phase. These observations resolve a significant concern about the origins of hysteresis in this material.

## Results

**Optoelectronic measurements.** We examined $CH_3NH_3PbI_3$ solar cells with two device architectures (see Methods for details): one showing limited $J$–$V$ hysteresis at room temperature (herein referred to as 'top cathode' cells—see Fig. 1a,c) and the other showing significant hysteresis in the photovoltaic performance at room temperature (referred to as 'bottom cathode' cells—see Fig. 1b,d).

To investigate the processes underlying hysteresis we examined the evolution of open circuit photovoltage ($V_{oc}$) with steady-state illumination, after preconditioning devices with a fixed bias voltage in the dark. This approach avoids the unnecessary complications introduced when analysing current–voltage sweeps, where both time and applied voltage are covariant. While monitoring the evolution of the $V_{oc}$ generated by the constant bias light (which we sometimes refer to as the background $V_{oc}$), a series of short (500 ns) laser pulses were simultaneously superimposed on the device to induce small perturbations in the photovoltage signal. Figure 2a shows a schematic of the experimental timeline for these 'transients of the transient' measurements. Analysis of the transient photovoltage perturbations gives information about changes in the movement and recombination kinetics of photogenerated charges as the background $V_{oc}$ evolves.

Hybrid perovskite solar cells are often preconditioned using an applied forward bias or illuminated open circuit conditions prior to measurement. This procedure changes the polarization of the device to a state in which higher efficiency values can sometimes be inferred from $J$–$V$ measurements than compared with short circuit or reverse bias preconditioning[1,3]. To explore this effect we have used two preconditions in this study: short circuit dark conditions ($V_{preset} = 0$ V) where the device is polarized by the built-in potential between the contacts ($V_{bi} \sim 0.9$–$1.3$)[41–43], or an applied forward bias ($V_{preset} = 1$ or $1.2$ V), which significantly reduces the potential, and thus the device polarization, between contacts. These two states form the starting conditions for the subsequent transient measurements.

Figure 3a shows the evolution of photovoltage of a top cathode solar cell, which had been preconditioned with a forward bias of $+1$ V in the dark for 1 min prior to switching the cell to open circuit and simultaneously turning the bias light on. After the initial development of the open circuit photovoltage to about 1 V (in less than 50 µs), there is then a small increase in $V_{oc}$ of approximately 10 mV over the course of the measurement. Throughout the measurement there was no significant change in the shape and time constants of the transient photovoltage (Fig. 3a,b). This is unsurprising since there is only a small change in the background $V_{oc}$ over the course of the measurement. Similarly stable results were also observed with this cell when it was preconditioned at short circuit ($V_{preset} = 0$ V) in the dark instead of $+1$ V (see Supplementary Fig. 2). These observations are consistent with the absence of significant hysteresis seen in the $J$–$V$ curves in Fig. 1c.

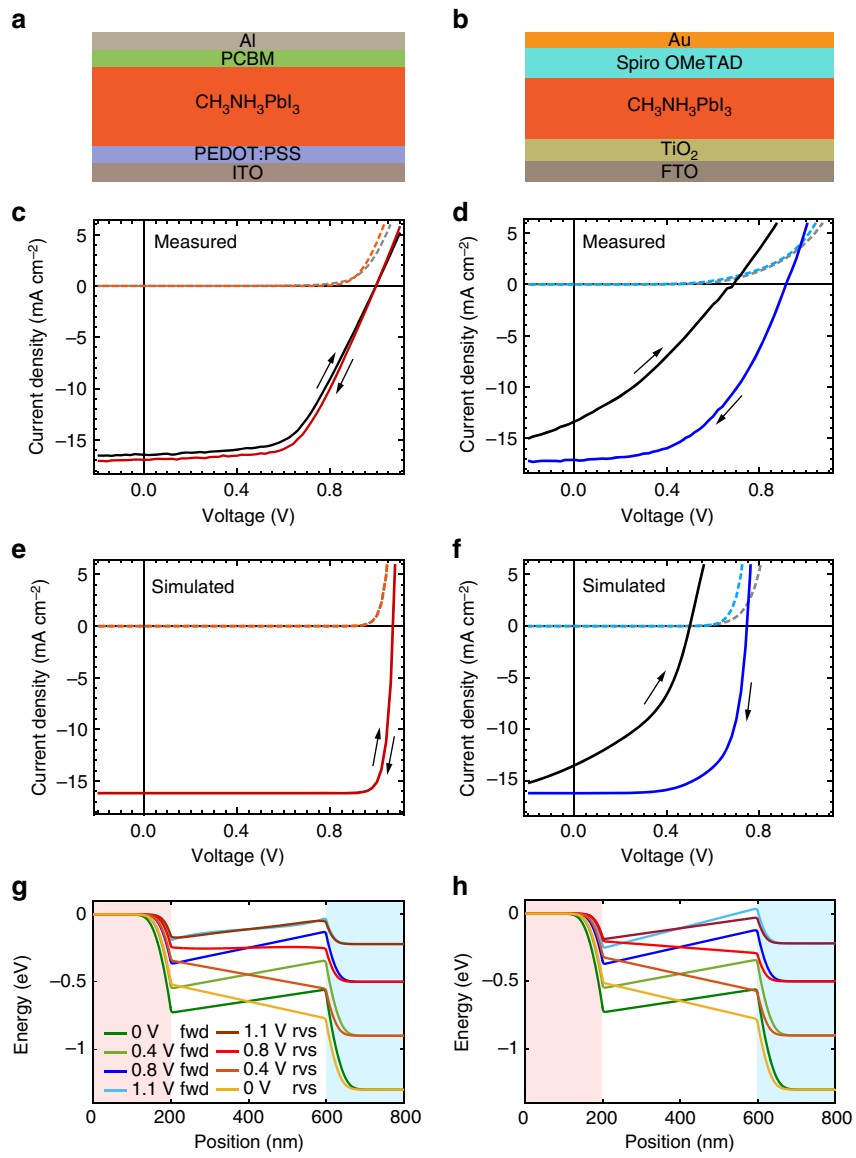

**Figure 1 | Measured and simulated device current–voltage characteristics.** (**a**) Top cathode and (**b**) bottom cathode perovskite solar cell device architecture stacks. Measured current–voltage curves in the dark and under 1 sun equivalent illumination scanned at approximately 40 mV s$^{-1}$ in the forward (reverse-to-forward bias) and reverse (forward-to-reverse bias) directions for the (**c**) top cathode cell, which showed a reverse scan (red curve) power conversion efficiency of 9.3% and a hysteresis index (HI) of 0.05 (defined in the Methods), and (**d**) the bottom cathode cell, which showed a reverse scan (blue curve) power conversion efficiency of 7.7% and HI = 1.71. Dashed orange and light blue curves show the reverse dark scan for top and bottom cathode devices, respectively. Black curves show the forward 1 sun scans, while dashed grey curves show the forward dark scan. The corresponding simulated current–voltage scans in each scan direction are shown for a p-i-n device structure with mobile ions, without (**e**) and with (**f**) recombination in the p- and n-type contact regions. The simulated values of HI were 0.00 and 1.84, respectively. The simulated scan protocol was similar to the experimental measurement at 40 mV s$^{-1}$ (see Methods for details). Simulated electrostatic potential profiles during forward (fwd) and reverse (rvs) J–V scans for the (**g**) top cathode and (**h**) bottom cathode device. The p- and n-type layers are indicated by pink and blue shaded regions, respectively.

In contrast, the bottom cathode device (Fig. 1b,d) exhibited very different behaviour upon preconditioning at different bias voltages. Figure 3c–f shows the $V_{oc}$ transients of the transient measurement performed on a bottom cathode solar cell, which showed significant hysteresis (Fig. 1c). Following a forward bias ( + 1 V) preconditioning step in the dark, the $V_{oc}$ declines steadily from around 850 to 720 mV in around 40 s under continuous illumination. Although the magnitudes of the small perturbation transient photovoltage decays decreased, the time constants (fitted using a double exponential function) during the measurement increased by a factor of approximately two over this time (Fig. 3c).

A zero-dimensional kinetic model of the device suggests that the factor of 2 increase in $\tau$ would correspond to a 36 mV increase in the photovoltage if the band gap and recombination reaction order were constant (see Supplementary Note 1 for details). This is clearly inconsistent with the observed decrease in $V_{oc}$, suggesting a more sophisticated model is required to describe device behaviour, as will be discussed later.

Figure 3e shows the evolution of the bottom cathode device photovoltage following preconditioning at short circuit (0 V) in the dark. In this case, the photovoltage rises by around 300 mV from 400 to 700 mV over 42 s. The simultaneous transient photovoltage signals in Fig. 3f exhibit anomalous behaviour: the

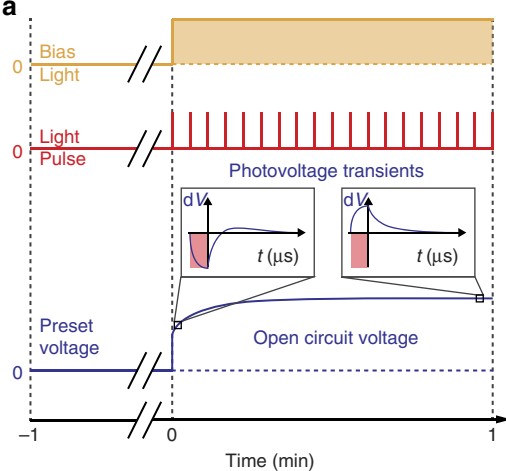

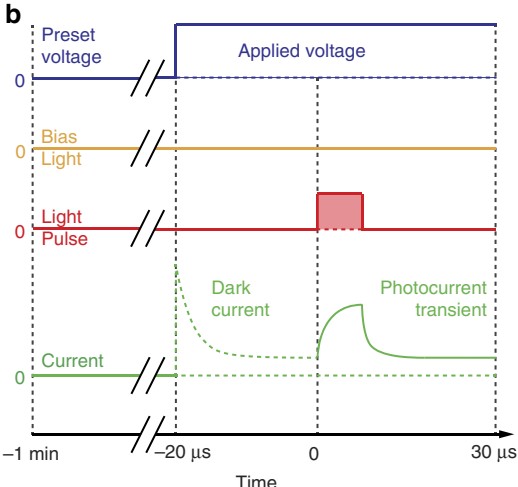

**Figure 2 | Experimental timelines for the optoelectronic transient measurements.** (**a**) Transients of the transient photovoltage measurement. The device is held in the dark (the bias light state is represented by the yellow line and shaded regions) at a preset voltage (dark blue line) for 1 min before being switched to open circuit with 1 sun equivalent illumination. During the $V_{oc}$ evolution the cell is pulsed with a 638 nm laser (red line and pink shaded regions) at 1 s intervals and the resulting photovoltage transients are recorded. (**b**) 'Step-Dwell-Probe' photocurrent transient measurement. The device is held in the dark for 1 min at a preset voltage (0 V in this study), then switched to an applied measurement voltage at which it is held for a further 20 µs, allowing the dark current (green dotted line) to stabilize. A 10 µs near-infrared LED pulse is then used to excite the cell and the transient photocurrent (solid green line) is recorded.

change in photovoltage during the light pulses at early times of the slow background $V_{oc}$ evolution is negative. By comparison, the tails of the transients are all positive, and the time constants for their decay towards quasi-equilibrium decrease by a factor of 2–4 over the course of the measurement (Fig. 3e). We would expect this change to correspond to a decrease in $V_{oc}$ of between 36 and 72 mV if all else were constant (Supplementary Note 1); this is also clearly inconsistent with the observed slow increase in voltage.

The negative deflection of the transient photovoltage measurements during the 500 ns transient light pulses indicates the existence of a positive internal current within the device (note that we define the sign of normal photocurrent to be negative, cf. Fig. 1). This additional positive current and associated negative displacement voltage at open circuit during the light pulse was

apparent for the first 20 s of the bias light exposure at open circuit. This implies that, during that time, a significant region of the internal electric field was opposite to that expected from the built-in potential, and was driving photogenerated carriers towards the wrong contacts. An accumulation of space charge must exist to generate this opposing E-field; below we show this is likely to be near the interfaces.

Taken at face value the results in Fig. 3a–f might suggest that ion migration, which could cause this charge accumulation, is present in the bottom cathode architecture but not in the top cathode devices (these do not show the negative photovoltage transients). To examine whether this is the case we also performed photocurrent transient measurements on both device types after stepping from a preconditioning bias voltage (0 V) to an applied forward bias near $V_{oc}$ (but below the built-in voltage) in the dark (see the experimental timeline in Fig. 2b). The dwell time was sufficiently short to allow the dark current to stabilize without significant ion migration occurring. This was verified by the observation that the photocurrent transients did not vary following dwell times of up to at least 500 µs. By stepping to an applied electrical bias near $V_{oc}$, instead of using a bias light to generate an open circuit photovoltage, the charge carrier transport direction can be probed without flooding the device with background photogenerated charges. The sign of the transient photocurrent reflects the direction of the dominant electric field in the cells.

The results of the measurements on the bottom cathode device are given in Fig. 4a. The control photocurrent transient measurement made at short circuit shows a negative photocurrent deflection, as expected. When the cell was switched to forward bias immediately prior to the transient measurement, we observe that the photocurrent transient is positive, consistent with the negative photovoltage transients observed for the bottom cathode device in Fig. 3f, which result from a positive internal photocurrent in the device. This observation confirms that there is an accumulation of space charge in the bottom cathode device causing an E-field opposing the built in potential.

Remarkably, when the measurement was repeated on the top cathode device, the photocurrent transient was similarly positive when the cell was stepped to a forward bias of 0.9 V (see Fig. 4b). This appears inconsistent with the purely positive photovoltage transients observed in Fig. 3b. The measurement indicates that, when there is no bias light flooding the device with photogenerated charge carriers, an E-field opposing the built-in potential initially exists. This is strong experimental evidence that there is also an accumulation of slow moving space charge in the top cathode architecture devices, despite the absence of significant hysteresis in the J–V scan at room temperature.

**Simulation of optoelectronic measurements.** As discussed in the introduction mobile ionic defects are widely thought to be intrinsically present in $CH_3NH_3PbI_3$. The accumulation of these charged defects at the contacts of devices due to the internal electric field within the perovskite layer has been used as a model to understand some hysteresis behaviour[1,6,7,16,18]. To test whether ion migration could explain the anomalous transient results observed here, we used a one-dimensional time-dependent drift-diffusion model that included a single, non-ionizing mobile ionic species, electrons, and holes (see Methods, Supplementary Tables 1–3 and Supplementary Note 1 for details). The focus of this study was not to realistically simulate all aspects of the devices, but to explore principal device behaviour.

For simplicity we simulated the devices as p-i-n structures in which the intrinsic perovskite layer is sandwiched by contacts approximated by p-type and n-type regions with identical band

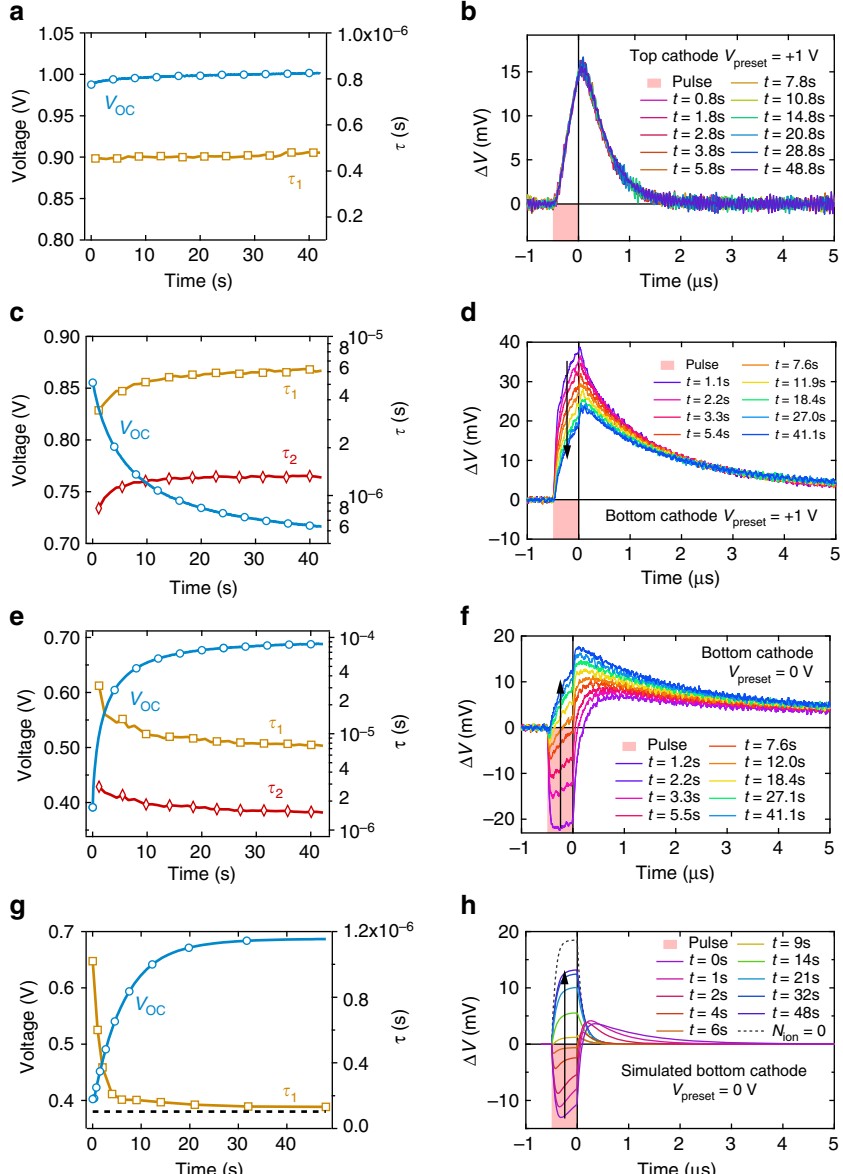

**Figure 3 | Transient measurements of the open circuit photovoltage evolution.** Slow timescale evolution of the open circuit voltage (left column, blue lines with circle markers) and corresponding small perturbation photovoltage transients (right column, where the shaded pink region represents the duration of the red laser pulse). The measurements were made using the protocol shown in Fig. 2a. The time constants $\tau_1$ (gold curve, square markers) and $\tau_2$ (red curve, diamond markers) from single or double exponential function fits to the tails of the photovoltage decay transients (see Methods) are plotted on the right-hand axis in the left column. (**a,b**) Top cathode device preconditioned in the dark at $+1$ V. (**c,d**) Bottom cathode device preconditioned in the dark at $+1$ V and (**e,f**) at 0 V. (**g,h**) Corresponding simulation of the photovoltage evolution and transient photovoltage measurements of a p-i-n device with mobile ions (with a background concentration of $N_{ion} = 10^{19}$ cm$^{-3}$) and high recombination in the p- and n- type regions with preconditioning of 0 V, cf Fig. 1d. The single exponential time constant from fits to the simulated photovoltage decays in (**h**) are also plotted (gold curve, square markers) in (**g**). For comparison, time constants from an otherwise identical simulation with no mobile ions are also included (black dashed lines).

gaps. The perovskite layer was set to contain a mean concentration of $10^{19}$ cm$^{-3}$ positively charged mobile ionic species (these could correspond to $I^-$ vacancies for example) with a corresponding uniform concentration of negative static ionic species. The top cathode architecture was assumed to have no recombination in the perovskite/contact interfacial regions. The only difference in simulating the bottom cathode device was the introduction of Shockley Read Hall (SRH) recombination in the contact materials to simulate recombination in these regions.

Figure 5 shows simulated energy level profiles and charge carrier densities for both top cathode and bottom cathode devices at the beginning (subscript 'i', solid lines) and end (subscript 'f',

dashed lines) of the $V_{oc}$ evolution and the Step-Dwell-Probe photocurrent simulations. The energy profiles include the potential of the conduction band ($E_{cb}$) and valence band ($E_{vb}$) as well as the quasi-Fermi levels of the electrons ($E_{Fn}$) and holes ($E_{Fp}$) relative to the potential of $E_{cb}$ at the left hand boundary (i.e. the anode). The charge carrier profiles show the density of free electrons ($n$), holes ($p$) and mobile ions ($a$). Figure 5a,b shows the simulated energy level profiles and charge carrier density distributions of a top cathode device after reaching equilibrium at short circuit in the dark. The simulated data for the bottom cathode device under these conditions is virtually identical (data not shown). At equilibrium mobile ionic charge screens the cell's

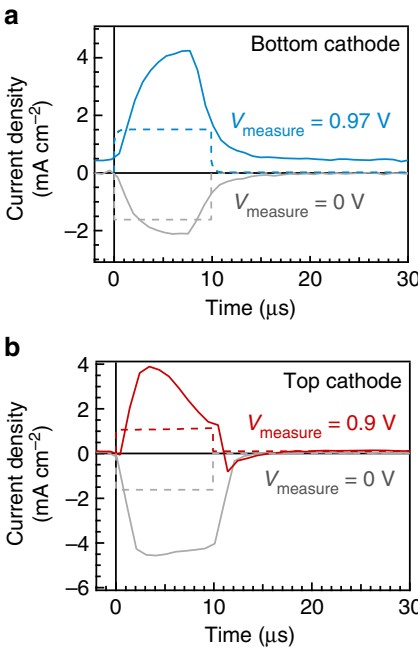

**Figure 4 | Step-to-Voltage transient photocurrent measurements.** The photocurrent transient measurements were taken using an 8 or 10 µs, 735 nm, pulse after a 20 µs dwell time following switching from 0 V to an applied forward bias in the dark ($V_{measure}$) following the protocol shown in Fig. 2b. (**a**) The solid blue curve shows the positive transient from a bottom cathode device where 0.97 V in the dark was applied following preconditioning at 0 V in the dark. (**b**) The red curve shows the positive photocurrent transient for a top cathode device with an applied measurement voltage of 0.9 V. In both cases, control photocurrent transients at short circuit are also presented (solid grey lines); these show negative photocurrents as expected. The dashed lines show the corresponding simulated photocurrent transients. Note that the rise and fall times of these transients are limited by the RC time constant of the device and the switch-on time of the LEDs used to create the light pulse.

built-in potential, created by the difference in Fermi energies of the p-type (pink shaded regions) and n-type (blue shaded regions) contact layers. This results in an accumulation of space charge at the perovskite/contact interfaces with an associated strong electric field, and a field-free region in the bulk of the device. The details of this distribution depend on the defect concentration, built-in potential and dielectric constant[18].

When the top cathode device is then illuminated at open circuit (simulating the experimental conditions in Fig. 2a), the initial build-up of photogenerated electrons and holes results in quasi-Fermi level splitting and the development of an open circuit potential, as expected. Immediately after illumination, the distribution of mobile ionic charges is as shown in Fig. 5b since they have not had time to move in response to the changed electric fields. In the presence of a photovoltage, this ionic charge distribution results in spatial electrostatic potential minima at the perovskite/contact interfaces, which we will refer to as 'valleys'. However, in the top cathode device, high concentrations of photogenerated electrons and holes rapidly redistribute to fill these valleys, resulting in screening of the initial E-field induced by ion accumulation (see Fig. 5c,d). Over a period of tens of seconds ions migrate away from the p-type region. This migration is a result of both the high ion concentration gradient at the interfaces and the reversal of the field direction, which now drives ions in the opposite direction to that during equilibration at short circuit (see Supplementary Fig. 3a). There is an accompanying redistribution of the electrons and holes throughout this time.

The key observation is that despite this ionic and electronic charge rearrangement, the change in $V_{oc}$ is very small (approximately 1 mV, see Supplementary Fig. 4), in agreement with the magnitude of the measured change seen in Fig. 3a and Supplementary Fig. 2.

Consistent with observation, hysteresis is not present in the simulated J–V curves of the top cathode device (Fig. 1e) despite the reversed E-fields in the perovskite region during the forward scan (Fig. 1g). The absence of recombination in the interfacial regions allows the accumulation of photogenerated charge carriers in the valleys that partially screen the ionic charge. This enables the efficient collection of carriers by diffusion as indicated by the gradients of the electron quasi-Fermi levels shown in Supplementary Fig. 5c. The simulated transients and J–V curves confirm that despite the slow redistribution of ionic defects within the material, significant hysteresis is not expected in the absence of surface or interfacial recombination.

When recombination in the contact layers is included, the simulation results replicate the slow evolution of the $V_{oc}$ following dark short circuit preconditioning seen in bottom cathode devices (Fig. 3e,g). From the simulated energy and charge distribution diagrams shown in Fig. 5e,f and Supplementary Fig. 3b (detail), it is apparent that potential valleys are also formed immediately following illumination. In this case, however, photogenerated electrons and holes collected in the valleys rapidly recombine due to high rates of recombination in the n- and p-type contact regions, and the electric field associated with ionic charge remains unscreened. Since the presence of the photovoltage partially negates the built-in potential, the concentration of ionic defects at the contacts decreases as the defects migrate away, until the E-field in the bulk of the cell is once again zero, consistent with recent scanning Kelvin probe observations[44]. At this point, the $V_{oc}$ reaches a plateau. The evolution of $V_{oc}$ during this time is dictated by (1) ion migration, (2) electronic charge rearrangement in response to ionic migration and (3) an increase in the concentration of photogenerated charge carriers due to redistribution away from fast SRH recombination centres in the contacts. An opposite process occurs for a bottom cathode device relaxing from forward bias preconditioning (Supplementary Fig. 6). These processes are also responsible for the difference in $V_{oc}$ between forward and reverse scans in the J–V curve (see Fig. 1f). Figure 1g,h and Supplementary Figs 5a,b show that the potential profiles and ion distributions are similar for both bottom cathode and top cathode simulations at low forward bias. However, recombination of carriers driven to the interfaces by the reversed E-field during the forward scan results in a significant loss of photocurrent (Supplementary Fig. 5d). The valleys are not present for the reverse scan, so charge collection is more efficient. The simulated J–Vs showed similar hysteresis indices (HIs) for the same scan speed and similar scan protocol (see Methods), with HIs of 0.00 and 1.84 compared with measured values measured for these devices of 0.05 and 1.71 for the top cathode and bottom cathode architectures, respectively. The effects of series resistance were not included accounting for the discrepancy in fill factor between experiment and simulation: the maximum power in the reverse scan would show a greater relative reduction than the forward scan due to higher current densities. We note that the degree of hysteresis observed in a J–V measurement is sensitive to the voltage scan rate[1], since this determines whether mobile ionic defects have time to react to the changing applied potential as has recently been simulated[18]. If the ratio of the scan rate to ionic mobility is very large the ions will appear 'frozen' in place during the scan, alternatively if the ratio goes to zero then ions will reach an equilibrium distribution for each voltage in both scan directions, and no hysteresis will be seen in either case.

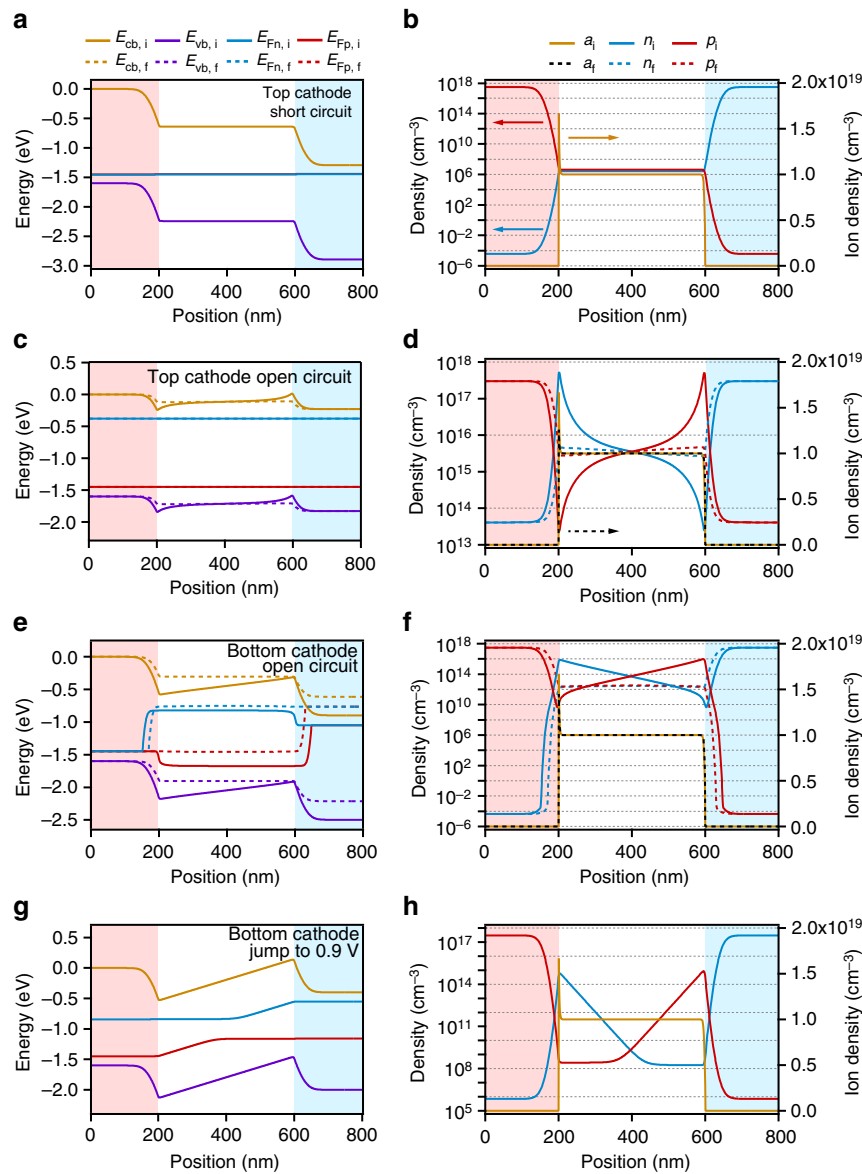

**Figure 5 | Simulated energy level and charge density profiles. (a,b)** Top cathode cell at short circuit in the dark (virtually identical results were simulated for the bottom cathode device, data not shown). Mobile ionic charge has drifted to completely screen the built-in potential between the p-type (pink shaded region) and n-type (blue shaded region) contact materials. **(c,d)** Top cathode cell at open circuit under illumination following short circuit in the dark. **(e,f)** Bottom cathode cell at open circuit under illumination after short circuit in the dark. **(g,h)** Top cathode cell after stepping to an applied forward bias of +0.9 V from short circuit in the dark. In **(c–f)** initial states (shown after 50 μs) are indicated by solid lines and subscript i while dashed lines and subscript f designate final states, after 50 s of ion migration under illumination at open circuit. Note that the energy scale is referenced to the potential of the conduction band at the left-hand boundary on the semiconductor energy scale to give the potential energy of an electron (the electrochemical scale has the opposite sign). $E_{cb}$ (gold) $E_{vb}$ (purple) $E_{Fn}$ (blue) and $E_{Fp}$ (red) refer to the energies of the conduction band, valence band, electron quasi-Fermi level and hole quasi-Fermi level respectively; $a$ (initial—gold solid line, final—black dashes), $n$ (blue) and $p$ (red) refer to the concentrations of mobile ionic charge, electrons and holes, respectively.

Our simulations also reproduce the anomalous transient photovoltage behaviour observed at early times during the $V_{oc}$ evolution (Fig. 3f,h and Supplementary Fig. 6 for the dark +1 V preconditioning). The negative transients are explained by the drift of the additional photogenerated charges to the valleys during the laser pulse (a positive internal photocurrent). Once the light pulse ends, the associated positive internal photocurrent stops and the excess electrons and holes then contribute to a positive deflection of the transient photovoltage in the normal way. This excess then decays away on a timescale reflecting the recombination kinetics of the device in its current state. The details underlying the photovoltage decay time constants are considerably more complex than the zero-dimensional model mentioned above (Supplementary Note 1), owing to the spatial separation of electrons, holes, and recombination regions. The simulation results in transient photovoltage decay time constants that decrease as the $V_{oc}$ increases, consistent with experimental trend (see Fig. 3e,g).

Simulation of the Step-Dwell-Probe transient photocurrent measurements is shown in Figs 4 and 5g,h. The presence or absence of interfacial recombination makes almost no difference to the simulated results since there are no photogenerated charge carriers prior to the pulse. Owing to the ionic charge distribution, positive photocurrent transients following the step to a forward

bias measurement voltage are observed in the simulations for both the top cathode and bottom cathode architectures, consistent with measured positive photocurrent transients in both cell types.

The simulations presented here contain only two features that differ from an ideal p-i-n solar cell: mobile ionic defects, which can result in a slow redistribution of charge in the perovskite layer, and for the bottom cathode cell, high rates of recombination in the p-type and n-type contact layers. These additions were sufficient to explain key features of the complex 'anomalous' behaviour observed in our experiments on timescales ranging from $10^{-8}$ to $10^2$ s with reasonable quantitative agreement.

For completeness, simulations with zero and low concentrations of mobile ions, with and without interfacial recombination, were tested. The results showed that mobile ion concentrations greater than $10^{17}$ cm$^{-3}$ are required to reproduce the observed behaviour (Supplementary Fig. 7). Other possible recombination schemes in the presence of ion migration were also investigated (Supplementary Note 3). These indicated that the recombination type in the contact regions did not need to be specifically SRH to reproduce the experimental results, and that high band-to-band recombination rates throughout the device or in the perovskite layer alone cannot reproduce the results (Supplementary Fig. 8). Simulation of significant recombination in only one contact layer yielded interesting results: it reproduced negative photovoltage transients at early times, but the $V_{oc}$ showed an initial increase which peaked after about 5 s followed by a decline to a plateau (Supplementary Fig. 8). This can be rationalized when one considers that the 'valley' at the p-type contact will initially accumulate electrons but will eventually discharge these for recombination at the n-type contact as the ions redistribute (see Supplementary Fig. 9). We have frequently observed biphasic evolution of the photovoltage with time in devices (e.g. Supplementary Fig. 10); this could be consistent with an asymmetry in the recombination rates in the two contact materials, where single-sided recombination is an extreme case of this possibility.

In summary, our study confirms that $CH_3NH_3PbI_3$ shows behaviour consistent with a mixed ionic/electronic conductor at room temperature. We have used optoelectronic photocurrent transient measurements with no bias light to demonstrate that the effects of ion migration can be observed in devices that exhibit minimal hysteresis in their J–V and transient $V_{oc}$ behaviour as well as those that show hysteresis. Simulation of the measurements shows that J–V hysteresis, slow timescale evolution of the $V_{oc}$ and negative photovoltage transient behaviour are reproduced by the combination of ion migration with high rates of recombination in the perovskite/contact interfaces. We note that this could also include recombination via pin holes, and could be partially attributable to the different substrates, morphology and perovskite deposition techniques used in processing each architecture[29]. Recent studies using alloyed hybrid perovskite preparations sometimes show relatively minimal hysteresis before ageing[45,46]. However, as well as standard hysteresis[47,48], in some cases inverted hysteresis is observed in these materials[39]. This suggests that recombination may be mediated by mobile defects themselves in addition to the interfaces, which is likely to be an interesting area for future investigation.

Our results provide experimental confirmation of the predictions from simulations by van Reenen et al.[16] that both ion migration and interfacial recombination are required for hysteresis to be observed. We also conclude that PCBM does indeed passivate interfacial recombination at the cathode relative to metal oxide contact materials, as suggested in previous studies[30,36,38]. However, our observations are not consistent with the hypothesis that PCBM reduces hysteresis by preventing

the diffusion of ionic defects along grain boundaries[30]. In addition to demonstrating the role of interfacial recombination in the presence of mobile ions in a semiconductor, our study demonstrates the viability of controlling the measureable consequences of this ion migration. This suggests the interesting possibility of exploiting these effects for other electronic applications where a memory of previous operating conditions would influence device behaviour[14,15].

## Methods

**Devices.** The planar bottom cathode devices had the following stack of layers: FTO glass/dense-TiO$_2$ ($\sim$50 nm)/CH$_3$NH$_3$PbI$_3$ ($\sim$300 nm)/Spiro-OMeTAD ($\sim$200 nm)/Au (80 nm) with an active area of 0.08 cm$^2$; the cells were prepared as described in ref. 3 (where FTO is fluorine-doped tin oxide; and Spiro-OMeTAD is N2,N2,N2′,N2′,N7,N7,N7′,N7′-octakis(4-methoxyphenyl)-9,9′-spirobi[9H-fluorene] -2,2′,7,7′-tetramine). The planar top cathode stack of layers was ITO glass/ PEDOT:PSS (30 nm)/CH3NH3PbI3 (300 nm)/PCBM (85 nm)/Ca (20 nm)/Al (100 nm) with an active area of 0.1 cm$^2$; the cells were prepared as described in ref. 34 (where ITO is indium-doped tin oxide; PEDOT:PSS is poly(3, 4-ethylenedioxythiophene):poly(styrenesulfonate and PCBM is phenyl-C61-butyric acid methyl ester). The key transient behaviour presented in this study was reproduced in all 10 working devices measured.

**Optoelectronic characterization.** Current–voltage sweeps of the devices were made using a Keithley 236 Source Measure Unit and a xenon lamp solar simulator with AM1.5G filters (Oriel Instruments). The illumination intensity was adjusted to be equivalent to 100 mW cm$^{-2}$ using a using a calibrated filtered Si photodiode (Osram BPW21). The J–V measurement protocol was as follows for both the light and the dark measurements. The cell was left at $-1$ V in the dark for approximately 30 s. In the case of the light measurements the solar simulator shutter was then opened. The applied voltage was then swept from $-1$ to $+1.2$ V at a rate of approximately 40 mV s$^{-1}$ (forward scan) and the current density measured, the optical shutter was then closed. The cells were then held at $+1.2$ V for a few seconds before the shutter was opened and the voltage swept back to $-1$ V at 40 mV s$^{-1}$ (reverse scan). As means to compare the degree of hysteresis between devices, we define a hysteresis index (HI = [$P_{max,r}/P_{max,f}$] $-1$) for a given scan rate in terms of the maximum power points on the reverse scan, $P_{max,\,r}$, and the forward scan, $P_{max,f}$. This differs slightly from the HI introduced by Kim and Park[49]. Top cathode devices showed minor variation in hysteresis (mean HI = 0.01 ± 0.03 from seven devices), while the bottom cathode devices showed greater hysteresis with more variation (mean HI = 4.5 from three devices, standard deviation 5.5, with minimum HI = 0.87).

Transient of the transient photovoltage measurements were made using a National Instruments USB-6361 data acquisition card to monitor the slow voltage transients (generated by a white bias light) and a Tektronix DPO5104B digital oscilloscope to monitor the fast voltage transients (generated by a red laser pulse). The two measurements were performed simultaneously (see Fig. 2a for the experimental timeline). The fast voltage transients were collected every second for approximately 45 s, averaging approximately 20 curves over approximately 200 ms. The white bias light was provided by an array of cool-white LEDs (Luxeonstar), calibrated to 1 sun equivalent with a silicon photodiode. The 500 ns laser pulse was provided by a digitally modulated Omicron PhoxX + 638 nm diode laser, with 100 Hz repetition rate. The laser spot size was expanded to cover the active pixel and the continuous wave intensity over the cell pixel area was approximately 550 mW cm$^{-2}$ during the pulse. The preconditioning bias was applied using the data acquisition card. The system was controlled by a custom Labview code. The measurement sequence is shown in Fig. 2a.

Single exponential small perturbation photovoltage decays ($\Delta V$) were seen in some top cathode devices, consistent with organic and dye sensitized solar cells. However, for many devices, most notably those with the bottom cathode architecture, the photovoltage transient decays could only be accurately fit using a bi-exponential function as has been reported previously[3,4,50,51]:

$$\Delta V = A_1 e^{-t/\tau_1} + A_2 e^{-t/\tau_2}, \tag{1}$$

where $A_1$ and $A_2$ are the amplitudes of the two components and $t$ is time. We have not yet confidently assigned the decay constants, $\tau_1$ and $\tau_2$ to specific physical phenomena although our observations suggests the biphasic decay is related to hysteresis and the probable asymmetry of recombination rates at the contacts (see Supplementary Figs 8 and 9). We note that the constants appear to vary proportionately and thus we use them to parameterize relative changes in the recombination lifetime.

The Step-Dwell-Probe photocurrent transients were made using light pulses from a 735 nm LED (Ledengin LZ1-00R300). The turn off time of the 735 nm LED was $\leq$200 ns as measured by a fast silicon diode and a GHz oscilloscope. Turn on time for the LED was $\leq$3 μs. The pulse intensity gave an absorbed photon flux approximately equivalent to 0.5 suns. Applied potential was supplied by a National Instrument USB-6251 multi-function DAQ board. Claimed output voltage slew

time is $20\,V\,\mu s^{-1}$ and settling time of less than $1\,\mu s$. Current was measured on the same USB-6251 with 16 bit resolution and $0.8\,\mu s$ per point.

**Drift-diffusion model.** A one-dimensional drift diffusion model was implemented to simulate the results using MATLAB's built-in partial differential equation solver for parabolic and elliptic equations (pdepe). The full details can be found in Supplementary Note 2, and Supplementary Tables 1–3 list the key simulation parameters, variables and constants. The code solves the continuity and Poisson's equations (Supplementary Note 2, equations (16)–(22)) for electrons, holes and positively charged mobile ionic defects (confined to the perovskite layer) and the electrostatic potential as a function of space and time. Note that neither the mobile nor the static ionic defects were modelled to induce doping effects that could liberate free electrons and holes in the conduction and valence bands, so the only role of the mobile defects in the simulation is to allow the distribution of charge in the device to change. Equilibrium and doping electron and hole densities were calculated using Boltzmann statistics (Supplementary Note 2, equations (23)–(25)).

To simulate current–voltage scans and current transients, a p-type/intrinsic/ n-type (p-i-n) structure device (Supplementary Fig. 11a) was used with an increasing or decreasing series of fixed potential difference boundary conditions (Supplementary Note 2, equations (29)–(34)) for each time step. The scan protocol used in the $J$–$V$ simulations was similar to that used experimentally (with a scan rate of $40\,mV\,s^{-1}$), and the final state of the device after the forward scan was used as the starting condition for reverse scan. The effects of series resistance were not included in the simulation.

Conventionally, the open circuit voltage is found by using Newton's method to solve for zero current at the boundaries. In order to accelerate calculation times and enable direct readout of the open circuit voltage, we used the method of image charges and devised a symmetric p-i-n (s-p-i-n) cell (Supplementary Fig. 11b). This approach enables simple Dirichlet boundary conditions to be employed by setting the potential at both boundaries equal to zero (Supplementary Note 2, equations (39) and (40)). Since charge densities in the two cells are a mirror of one another, the current and electric field at the mid-point of the device go to zero. However, the potential at the mid-point is free to change in response to changes in carrier density profiles in accordance with Poisson's equation. Provided that the depletion widths at the intrinsic/doped layer interfaces are significantly less than the layer thicknesses themselves, the doping concentrations in the n- and p-type layers generate the built-in voltage in the device. The $V_{oc}$ of the cell is obtained by taking the difference in electron quasi-Fermi energy, $E_{Fn}$ at the mid-point of the device (cathode) and the hole quasi-Fermi energy, $E_{Fp}$, at the left-hand boundary (anode) (Supplementary Note 2, equation (26)). For clarity, band diagram and charge density figures included herein for the s-p-i-n model only show the left-hand half of the device.

The model used a mesh with a linear grid $x$ spacing of $0.67\,nm$ per point (1,200 and 2,400 points for p-i-n and s-p-i-n models, respectively), which is marginally larger than a $CH_3NH_3PbI_3$ lattice cage width of $0.63\,nm$ (ref. 52). Examples of data calculated with other grid spacings are shown in Supplementary Fig. 12; we chose $0.67\,nm$ as a compromise between numerical accuracy and calculation time.

To further accelerate calculation times, generation in the active layer was uniform, and adjusted to yield an absorbed photon flux equivalent to $16\,mA\,cm^{-2}$ ($2.5\times10^{21}\,cm^{-3}\,s^{-1}$). Voltage transients and current transients were taken using fluxes of 16 and $3.2\,mA\,cm^{-2}$, respectively. In simulating the top cathode device, SRH recombination was switched off, while for the bottom cathode devices, an SRH recombination was implemented in the contact regions with a time constant of $2\times10^{-15}\,s$. The band-to-band recombination coefficient was chosen ($10^{-10}\,cm^{-3}\,s^{-1}$) such that the time constants for the transient photovoltage decays in the top cathode device were similar those observed experimentally (Fig. 3b). While unrealistic, including SRH recombination throughout the thickness of the contact regions rather than solely at the interfaces circumvented numerical inaccuracies resulting from the combination of a high potential gradients and high recombination currents at the perovskite contact interfaces. Using this method approximately 75% of the total SRH recombination takes place within the first 10 nm of the contact region (e.g. between 190 and 200 nm in the p-type region), yet the same $V_{oc}$ was obtainable using an order of magnitude higher recombination time constant than when a 5 nm recombination layer was used at the interface.

In the bottom cathode device, both the SRH time constant and trap energies determine the overall rate of SRH recombination (Supplementary Note 2, equation (28)). In the absence of data regarding possible trap energies, the levels were chosen arbitrarily to be shallow (0.2 eV below the conduction band and 0.2 eV above the valence band for the n and p-type regions, respectively). Since deeper trap energies combined with a lower recombination time constant yield similar behaviour, our simulations do not provide an estimate for the SRH time constant in real devices. The combination of SRH trap energy and time constant were chosen to result in a steady-state $V_{oc}$ value approximately corresponding to that observed in the bottom cathode device.

The timescales and magnitudes of the $V_{oc}$ transients were sensitive to choice of built-in voltage (which was not adjusted for the different architectures), SRH recombination rates, initial ionic charge density and ion mobility.

**Data availability.** The data that support the findings of this study are available from dataexss@imperial.ac.uk upon request.

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

## Acknowledgements

We thank Diego Alonso Alverez and Andrew McMahon for consultation on the drift diffusion simulation, and Davide Moia, Joel Troughton and Matthew Carnie for helpful discussions on devices and measurements. We thank James Durrant for use of facilities and the WAG funded Sêr Cymru Solar project for funding. AMT thanks the Imperial College Junior Research Fellowship Scheme for support. We are grateful to the U.K. Engineering and Physical Sciences Research Council for financial support (grants EP/J002305/1, EP/M023532/1, EP/I019278/1, EP/M025020/1, EP/G037515/1 and EP/M014797/1).

## Author contributions

P.R.F.B. and B.C.O.R. designed the study. A.M.T., P.C. and B.C.O.R. performed the experimental measurements. A.M.T. and B.C.O.R. designed and built the experimental setups. P.C. and P.R.F.B. performed the simulations. D.B. and X.L. fabricated the samples. All authors contributed to the interpretation of results and preparation of the manuscript.

## Additional information

**Competing financial interests:** The authors declare no competing financial interests.

