## [Peer Review File · Nature Communications]

Reviewers' comments:

Reviewer #1 (Remarks to the Author):

This paper describes measurements of the current-voltage characteristics and transient photovoltage and photocurrent in a perovskite solar cell caused by a sudden change in the applied bias followed by a light pulse. Cells with top cathode and bottom cathode configurations are compared. A drift diffusion, DD, model that allows for mobile ions is solved for comparison with the measurements.

The measurements are novel and clever. Whilst the model is a standard DD model it is applied in a novel context. The authors argue that their results can be understood by assuming that hysteresis requires a combination of ionic charge buildup and photogenerated charge recombination near the contacts. This conclusion is very interesting but needs to be more clearly demonstrated. If the concerns I describe below, that arise from a lack of clarity in the presentation of the model and results can be addressed then the paper could be accepted for publication.

The authors argue that the only way that hysteresis can be understood is from recombination at interfacial traps. The authors state that 'These observations undermine the viability of the ionic diffusion model for explaining hysteresis because the effect appears to be controlled by the interface properties rather than processes in the bulk of the active layer'. This statement is not obviously consistent with the notion published by the authors elsewhere that hysteresis can be caused by ionic charge build up at the interfaces.

There are many places where the discussion is not clear. An example is the lack of discussion of the modelling results in Figs 1e,f. There is no clear discussion on the effects of preconditioning. Undefined terms are used such as 'background Voc'. Unsubstantiated statements are made such as 'The dwell time was sufficiently short to allow the dark current to stabilise without significant ion migration occurring'.

Information is missing from the model description. Some of the parameter values in the model, e.g. the ion concentration and mesh width, are justified yet the model predictions depend sensitively on these values. It should also be stated how the displacement current is calculated in this model. A rationale should be provided for the amount of recombination assumed for the two cells.

Reviewer #2 (Remarks to the Author):

This paper reports a useful characterization using transient measurement to evaluate the role of the ionic migration and contact recombinations on the hysteresis effect of planar-type perovskite solar cells. To explain this, two type of devices were realized and measured using transient voltage measurements using preconditioning conditions. The experimental results were also associated to the simulation ones showing that the ionic migration is not the main factor leading the hysteresis effect but also contact recombinations have to be present to show this effect on the IV measurements.

The manuscript is well written with a clear demonstration of the thesis explained in the introduction. The paper is also novel and well structured with a detailed description of the experimental procedures and the characterization ones. I feel this the paper deserves publication in Nature Communication.

However, before the manuscript can be accepted for publication the authors need to address the following points.

1) the hysteresis effects showed in fig. 1a-b for top and bottom cathodes are remarked on a batch of eight devices. The author should show statistical results about the hysteresis index for both type of devices.

2) Why the simulated IV curves were realized using different scan rate (70mV/s) with respect to measured IV (40mV/s)? In literature was reported that the scan rate has a remarkable impact on the hysteresis effect especially for planar PSC cell. The author should explain this point.

3) To help the readers, the author should report the conversion efficiency of the cells showed in fig. 1a-b (also in the figure caption). Furthermore, the cells should have an efficiency lower than 10% related to low Voc Values with respect to the state of art. The author can explain if the results obtained for transient characterization are dependent from the initial efficiency and Voc values of the device. Could the author show further measurement details using high-efficiency solar cell?

4) The author reports this sentence about the preconditioning (pag.4 line 127): "This preconditioning step is analogous to the forward bias (or illuminated open circuit conditions) often applied to perovskite solar cells prior to measuring a current-voltage curve". Is the method used to measure the cells in fig. 1a-b? The author should clarify this point.

5) Caption of the Fig.5 is too long. The author should describe the reported comments in the Supplementary information.

Reviewer #3 (Remarks to the Author):

The hysteresis phenomena are unique and interesting in the perovskite solar cells. There are many reports on the hysteresis behaviors: modulating hysteresis and their causes. However, the causes for the hysteresis are still unclear. The authors investigate the ionic migration and hysteresis by measuring and simulating the transient optoelectronic properties of two perovskite solar cells with top and bottom cathode, respectively. They claim that ionic migration can be observed in hysteresis-free devices through optoelectronic photocurrent transient measurements. I don't recommend the manuscript to publish in Nature Communication at current stage. Here are my comments:

1. The authors design two device architectures with top and bottom cathode in their experiments. The device structure might have effects on the hysteresis, but not the only reason. There are many reports on the hysteresis-reduce and even hysteresis-free perovskite solar cells with traditional device architecture and bottom cathode (Li X, Science, 353,6294, 2016; Jeon NJ, Nature, 517, 476, 2015; Bi, Sci. Adv. 2016, 2:e1501170; Son DY, Nature Energy, 16081, 2016, DOI: 10.1038/NENERGY.2016.81; Saliba M, Energy Environ. Sci. 2016, DOI: 10.1039/c5ee03874j). So, it is unsuitable to reveal the causes for hysteresis by only investigating the above two device architecture.
2. The authors use pulse laser for optoelectronic photocurrent transient measurements. As reported previously (Lian L. Scientific Reports, 2015, 5, 16563), the optoelectronic response depends on the parameter of laser and quality of perovskite sensitively. The authors should provide more details about the pulse laser beam, such as the spot size, wavelength, position of the solar cells. They also should provide some details on the perovskite layer in the manuscript.
3. The authors use a p-i-n model to simulate the perovskite solar cells. However, the structure of the perovskite differs evidently from a traditional p-i-n junction. As they stated in their manuscript, the perovskite is a mixed ionic/electronic conductor, especially under illumination (also see Fig. 2). The perovskite film cannot be treated as an intrinsic layer. As shown in Fig. 1, the simulation deviates evidently from the experimental results. So, the p-i-n junction is not suitable for the perovskite solar cells.

Reviewer #1 (Remarks to the Author):

This paper describes measurements of the current-voltage characteristics and transient photovoltage and photocurrent in a perovskite solar cell caused by a sudden change in the applied bias followed by a light pulse. Cells with top cathode and bottom cathode configurations are compared. A drift diffusion, DD, model that allows for mobile ions is solved for comparison with the measurements.

The measurements are novel and clever. Whilst the model is a standard DD model it is applied in a novel context. The authors argue that their results can be understood by assuming that hysteresis requires a combination of ionic charge buildup and photogenerated charge recombination near the contacts. This conclusion is very interesting but needs to be more clearly demonstrated. If the concerns I describe below, that arise from a lack of clarity in the presentation of the model and results can be addressed then the paper could be accepted for publication.

The authors argue that the only way that hysteresis can be understood is from recombination at interfacial traps. The authors state that 'These observations undermine the viability of the ionic diffusion model for explaining hysteresis because the effect appears to be controlled by the interface properties rather than processes in the bulk of the active layer'. This statement is not obviously consistent with the notion published by the authors elsewhere that hysteresis can be caused by ionic charge build up at the interfaces.

We agree that this statement appears ambiguous and apparently self-contradictory. However, our intended meaning was that the contact material appears to control the degree of hysteresis observed, and not the perovskite phase itself. Even though the ionic charge build up at the interfaces can be considered an interfacial effect it is contained within the bulk of the perovskite for the purposes of our model.

To reduce this ambiguity, we have replaced the sentence with:

“These observations undermine the viability of the ionic diffusion model for explaining hysteresis because the effect appears to be controlled by the contact material properties rather than the accumulation of mobile ions in perovskite layer at the interfaces.”

There are many places where the discussion is not clear. An example is the lack of discussion of the modelling results in Figs 1e,f.

There was very little discussion of the simulated current voltage curves, and additionally the brief mention of these results in the text was miss-referenced to figure 1b instead of figure 1e. This has been updated and brief discussion of these simulations added. We have also added figures S5 to the supporting information showing the evolution of the ionic and electronic distribution during a *J-V*

curve in each scan direction, as well as additional panels to figure 1 showing the evolution of the electrostatic potential during the J - V curves. These add further insight to the underlying mechanism of hysteresis.

We have also updated the of the results section discussing the simulations in numerous places (highlighted) to improve the clarity of the discussion.

There is no clear discussion on the effects of preconditioning.

This is a valid criticism, we realise that although we included results on the effect of preconditioning for the transient simulations in the supporting information, we allocated virtually no space in the main text to analysing these observations.

The effects of preconditioning on the J - V curves of devices displaying hysteresis has been discussed in considerable detail by Tress et al. [10.1039/c4ee03664f] which we refer the readers to. We focussed our additional discussion primarily of the effects of preconditioning on the transient measurements. The following paragraph now motivates the results:

“Hybrid perovskite solar cells are often preconditioned using an applied forward bias or illuminated open circuit conditions prior to measurement. This procedure changes the polarization of the device to a state in which higher efficiency values can sometimes be inferred from J - V measurements than following short circuit or reverse bias preconditioning.^{1,3} To explore this effect we have used two preconditions (V_{preset}) in this study: short circuit dark conditions where the device will be polarized by the built-in potential between the contacts (thought to around $V_{bi} \sim 0.9 - 1.3$)⁴²⁻⁴⁴, or a applied forward bias of either 1 or 1.2 V, which significantly reduces the potential between contacts, and thus the device polarisation. These two states form the starting conditions for the subsequent transient measurements.”

We have also added discussion to the simulation results section on the effects of preconditioning.

Undefined terms are used such as 'background Voc'.

The term 'background Voc' has been defined in the main text: “While monitoring the evolution of the V_{oc} generated by a constant bias light (sometimes referred to as the background V_{oc}), ...”

Unsubstantiated statements are made such as 'The dwell time was sufficiently short to allow the dark current to stabilise without significant ion migration occurring'.

Although we had substantiated this statement in the lab, we did not describe how in the manuscript. We tested the measurement at a wide variety of dwell (delay) times following the step to the

measurement voltage after the capacitive component of the dark current stabilised (see examples in the figures below). We observed that the resulting photocurrent transient is identical and independent of the delay time until around 2 ms. We interpret this to indicate that the state of the devices have not changed significantly within the dwell time (consistent with no significant ion migration).

Left figure. Transient current measurements made on the bottom cathode device in the main text for two different dwell times: 40 μs (blue) and 120 μs (red). The laser pulse began at 0 μs . In both cases the capacitive charging current has decayed after approximately 15 μs , but the photocurrent transients at time = 0 look the same. **Right figure.** The photocurrent transient for the top cathode device for a series of different dwell times (shown in the legend), the capacitive charging is not shown in this example.

Our simulations (using the literature diffusion coefficients used in the model) also verified negligible change in the electrostatic distribution within the cell due to ionic charge on the dwell timescale used.

We have added an additional sentence to the main text: 'The dwell time was sufficiently short to allow the dark current to stabilise without significant ion migration occurring. This was verified by the observation that the photocurrent transient did not vary for dwell times of at least 500 μs .'

Information is missing from the model description. Some of the parameter values in the model, e.g. the ion concentration and mesh width, are justified yet the model predictions depend sensitively on these values. It should also be stated how the displacement current is calculated in this model. A rationale should be provided for the amount of recombination assumed for the two cells.

We have added a figure to the supplementary information (figure S12) justifying our choice of mesh spacing in the model, and a brief description to the main text.

The dependence of the simulation behaviour on ionic defect concentration dependence was shown in the figure S7 and also mentioned in the main text.

The relationship that we used to evaluate the displacement current is now given in Note S3.

The recombination rate constants were chosen for approximate consistency with the observed transients and V_{oc} values as is now described more clearly in methods of the main text.

Reviewer #2 (Remarks to the Author):

This paper reports a useful characterization using transient measurement to evaluate the role of the ionic migration and contact recombinations on the hysteresis effect of planar-type perovskite solar cells. To explain this, two type of devices were realized and measured using transient voltage measurements using preconditioning conditions. The experimental results were also associated to the simulation ones showing that the ionic migration is not the main factor leading the hysteresis effect but also contact recombinations have to be present to show this effect on the IV measurements.

The manuscript is well written with a clear demonstration of the thesis explained in the introduction. The paper is also novel and well structured with a detailed description of the experimental procedures and the characterization ones. I feel this the paper deserves publication in Nature Communication.

However, before the manuscript can be accepted for publication the authors need to address the following points.

1) the hysteresis effects showed in fig. 1a-b for top and bottom cathodes are remarked on a batch of eight devices. The author should show statistical results about the hysteresis index for both type of devices.

We have defined a hysteresis index as $HI = (P_{max_reverse}/P_{max_forward}) - 1$, now described in the methods. This is a little different to the definition used by Park et al. [JPCL, 10.1021/jz501392m] as we felt their definition, which involves a comparison of photocurrents at 0.8 V, was not a sufficiently general metric to be used here. The mean and standard deviation of the hysteresis index for the devices examined are quoted in the methods. The hysteresis indices are also now compared with the simulated values, in the simulation results section.

2) Why the simulated IV curves were realized using different scan rate (70mV/s) with respect to measured IV (40mV/s)?

We have repeated the simulations with a 40 mV/s scan rate, and these are now presented along with the calculated electrostatic potential, ion concentration and quasi-Fermi level profiles within the device during the forward and backwards scans (figure 1g, 1h and figure S5). We believe these profiles add considerable insight to the mechanism underlying $J-V$ hysteresis. There is good

agreement between the simulated and measured results, in which the hysteresis indices are reproduced.

In literature was reported that the scan rate has a remarkable impact on the hysteresis effect especially for planar PSC cell. The author should explain this point.

We agree that scan rate has a significant influence on degree of hysteresis observed in devices, and point out that this has been discussed by Tress et al. (10.1039/c4ee03664f) and simulated by Richardson et al. (10.1039/C5EE02740C). One of our motivations for examining the evolution of V_{oc} with time in this work was to avoid focussing on JV scans and the associated complication of analysing the effects of simultaneous changes in time and applied voltage. We have added a very brief discussion on the effects of scan rate to the main text.

3) To help the readers, the author should report the conversion efficiency of the cells showed in fig. 1a-b (also in the figure caption). Furthermore, the cells should have an efficiency lower than 10% related to low V_{oc} values with respect to the state of art. The author can explain if the results obtained for transient characterization are dependent from the initial efficiency and V_{oc} values of the device. Could the author show further measurement details using high-efficiency solar cell?

To address these points, we fabricated additional top-cathode devices, some of these showed higher V_{oc} values. We have replaced the results from the top-cathode device with a new example of a device fabricated in the same way, but showing a higher efficiency and voltage (approximately 10%). The various transient measurements repeated on this device indicated the same behaviour as previously (these have been updated in the manuscript).

We have reported the conversion efficiencies that would be inferred from the cells shown in figs 1a and 1b. We note however, that these values are not of central relevance to the message of the paper, and quoting efficiency values is somewhat questionable in a class of devices which still show such poor stability on a time scale of days or weeks. There are many examples of high efficiency cells in the literature, and these are almost always measured within the first few hours following fabrication/illumination. These performances are generally ephemeral, and stability data presented in the same papers is almost without exception starting from a lower baseline. We feel that measuring devices which are representative of the state following the initial drop in performance is probably of most relevance to the field since the state of the devices is likely to be more constant over a sustained period of measurement.

We specifically chose model systems with simple planar architectures to reduce the complexity of our system and for ease of modelling. The V_{oc} values of our devices are comparable to similar architectures published in the literature.

The example we now show for our 'top cathode' architecture device has a V_{oc} of about 1 V (the previous example was around 0.8V) this is now slightly above many 'baseline' examples using this architecture with similar contact materials of around 840 – 880 mV seen for example in Seo et al. EES, 2014 (10.1039/C4EE01216J), Jiang et al. J. Mater. Chem. A 2016 (10.1039/C5TA09231K), and Shahbazi et al. RSC Adv. 2016 (10.1039/C6RA11936K). There are however examples in the literature where improvements to the V_{oc} are demonstrated by modifying the interfacial properties. For example, adding excess PbI_2 to the layer can push the V_{oc} to just beyond 1V in these architectures which also reduces hysteresis (Chang et al. J. Mater. Chem. A, 2016, 10.1039/C6TA00679E), consistent with our proposed mechanism. However, in an unrelated study we have found that a PbI_2 excess did always result in either reduced hysteresis or improved performance (the difference probably depend on some variation in the processing details).

For our planar 'bottom cat' architecture devices with similar interfaces, the V_{oc} around 0.9 V we observed is similar to typical values for solution processed devices found in the literature between 830 and 920 mV (Chen et al. JACS, 2014, 10.1021/ja411509g), although we note that a V_{oc} of 1070 mV was achieved using vapour deposited films (Liu et al, Nature, 2013, 10.1038/nature12509), where more defect-free surfaces were achieved.

4) The author reports this sentence about the preconditioning (pag.4 line 127): "This preconditioning step is analogous to the forward bias (or illuminated open circuit conditions) often applied to perovskite solar cells prior to measuring a current-voltage curve". Is the method used to measure the cells in fig.1a-b? The author should clarify this point.

We have attempted to clarify our discussion of preconditioning in the main text as discussed in our response to Reviewer #1.

We have also more clearly stated the protocols used to measure the devices and the preconditions used for the simulations in the methods.

5) Caption of the Fig.5 is too long. The author should describe the reported comments in the Supplementary information.

We have significantly reduced the length of the caption.

Reviewer #3 (Remarks to the Author)

The hysteresis phenomena are unique and interesting in the perovskite solar cells. There are many reports on the hysteresis behaviors: modulating hysteresis and their causes. However, the causes for the hysteresis are still unclear. The authors investigate the ionic migration and hysteresis by measuring and simulating the transient optoelectronic properties of two perovskite solar cells with

top and bottom cathode, respectively. They claim that ionic migration can be observed in hysteresis-free devices through optoelectronic photocurrent transient measurements. I don't recommend the manuscript to publish in Nature Communication at current stage. Here are my comments:

1. The authors design two device architectures with top and bottom cathode in their experiments. The device structure might have effects on the hysteresis, but not the only reason. There are many reports on the hysteresis-reduce and even hysteresis-free perovskite solar cells with traditional device architecture and bottom cathode (Li X, Science, 353,6294, 2016; Jeon NJ, Nature, 517, 476, 2015; Bi, Sci. Adv. 2016, 2:e1501170; Son DY, Nature Energy, 16081, 2016, DOI: 10.1038/NENERGY.2016.81; Saliba M, Energy Environ. Sci. 2016, DOI: 10.1039/c5ee03874j). So, it is unsuitable to reveal the causes for hysteresis by only investigating the above two device architecture.

A central aim of our study was to investigate why some devices exhibit hysteresis while others do not, and to probe if ionic migration is present in both cases. To answer this question we required devices that did, and did not, show significant hysteresis at room temperature. As long as the mechanism underlying hysteresis is common to devices which display it, the details of the architecture are of somewhat secondary importance. We believe the mechanism underlying hysteresis, when it is observed, is similar for devices made with $\text{CH}_3\text{NH}_3\text{PbI}_3$ and thus our method is suitable for revealing the causes of hysteresis.

We agree that some, but not all, all of the devices described in the cited studies do not show much hysteresis. However, the active material used in these solar cells is chemically different in all but one of these reports to the $\text{CH}_3\text{NH}_3\text{PbI}_3$ that we used to demonstrate the hysteresis effects. The alloyed (mixed) perovskites in the papers cited could be expected to change both the nature of any ionic/defect migration and also the nature of interfacial recombination effects. Furthermore, all cases cited by the reviewer have a different architecture (using mesoporous TiO_2 on the substrate) in contrast to our planar devices. Mesoporous TiO_2 modifies the quality of the interface with the perovskite phase relative to a planar TiO_2 interface.

For our study we intentionally chose the simplest possible planar architectures in order to reduce the number of possible variables introduced by more complex active layer compositions and mesostructuring. This allowed us to more easily isolate specific effects. The key comparison required a method to change the magnitude of surface recombination in different devices. We achieved this by changing TiO_2 and spiro contacts for PCBM and PEDOT:PSS. There are undoubtedly alternative modifications to the device fabrication details that also result in a significant change in hysteresis as suggested by the reviewer. We have already provided an alternative example in the Supplementary Information where PCBM is replaced with ZnO to increase hysteresis in the 'inverted' (top-cathode) architecture.

Below is a summary of the devices used in the papers quoted by Reviewer #3.

Li et al., Science, 353, 6294, 2016 (10.1126/science.aaf8060) used an alloyed phase $\text{FA}_{0.81}\text{MA}_{0.15}\text{PbI}_{2.51}\text{Br}_{0.45}$ where MA is CH_3NH_3^+ , and FA is $\text{CH}_3(\text{NH}_2)_2^+$ for their active layer, this was deposited on mesoporous TiO_2 in contrast to the planar TiO_2 architectures that we used.

Jeon NJ, Nature, 517, 476, 2015 (10.1038/nature14133) also used an alloyed phase $(\text{FAPbI}_3)_{1-x}(\text{MAPbBr}_3)_x$ deposited on mesoporous TiO_2 , and also observed indications of some mild hysteresis (much less than MAPbI_3 in a similar architecture).

Bi et al., Sci. Adv. 2016, 2:e1501170 (10.1126/sciadv.150117) also used an alloyed phase $\text{FA}_{1-x}\text{MA}_x\text{Pb}(\text{I}_{1-y}\text{Br}_y)_3$ on mesoporous TiO_2 they minimised hysteresis in their devices by adding a carefully controlled excess of PbI_2 , which might also influence the concentration of ionic defects or interfacial recombination.

Saliba M et al. (Energy Environ. Sci. 2016, (10.1039/c5ee03874j) use an alloyed perovskite phase containing: $\text{Cs}_x(\text{MA}_{0.17}\text{FA}_{0.83})_{(100-x)}\text{Pb}(\text{I}_{0.83}\text{Br}_{0.17})_3$. These devices do show some signs of hysteresis (although this can be reversed behaviour), and an initial study has recently been published on this subject by Tress et al. from the same group in Advanced Energy Materials (10.1002/aenm.201600396). We believe that the model we have developed will be suitable for unravelling this behaviour, however, the unusual behaviour is related to the details of the recombination processes near the interfaces which are more complex than we have discussed in the present study.

We now briefly mention the hysteresis behaviour of these mixed phase materials and highlight them as an area for future research efforts in the concluding remarks.

Son et al., Nature Energy, 16081, 2016, (10.1038/NENERGY.2016.81) is the only study listed to examine MAPbI_3 although they also used mesoporous TiO_2 substrates. They showed that by adding an excess of MAI they could reduce non radiative recombination at grain boundaries/interfaces and the resulting devices showed minimal hysteresis. These empirical observations are consistent with our model/explanation of why no hysteresis would be expected in this situation. This work is now cited in the introduction.

2. The authors use pulse laser for optoelectronic photocurrent transient measurements. As reported previously (Lian L. Scientific Reports, 2015, 5, 16563), the optoelectronic response depends on the parameter of laser and quality of perovskite sensitively.

In this interesting study by Lian et al. (10.1038/srep16563) laser pulses were used to make lateral photocurrent transient measurements on perovskite crystals and films which showed similar response time scales (now referred to in the manuscript). The responsivity of these devices was dependent on laser power. It is likely that this varying responsivity in the lateral devices is another manifestation of the hysteresis effects seen in solar cells where, for example, steady state short circuit photocurrent can depend non-linearly on illumination intensity in devices showing significant

hysteresis. What is new about our measurements is that we combined transient optoelectronic measurements on two entirely different timescales for perovskite solar cells, and used a carefully controlled set of device switching and prebiasing protocols. The transient measurements are in the small perturbation regime where the important factor is the shape and time scale of the response rather than its absolute magnitude.

The authors should provide more details about the purl laser beam, such as the spot size, wavelength, position of the solar cells.

We did provide details on the laser pulses in the methods section of the manuscript: “The 500 ns laser pulse was provided by a digitally-modulated Omicron PhoxX+ 638 nm diode laser, with 100 Hz repetition rate and a continuous wave intensity over the cell pixel area of approximately 550 mW cm⁻² during the pulse.” In a revised manuscript we now explicitly state that the spot size was expanded to be larger than the pixel area.

They also should provide some details on the perovskite layer in the manuscript.

It is not entirely clear what is required here beyond their composition and fabrication details of the films. We note that our study is focussed on the device behaviour and simulation rather than the film properties. More detailed physical characterisation of the materials used to fabricate our devices can be found in the references cited in the paper.

3. The authors use a p-i-n model to simulate the perovskite solar cells. However, the structure of the perovskite differs evidently from a traditional p-i-n junction. As they stated in their manuscript, the perovskite is a mixed ionic/electronic conductor, especially under illumination (also see Fig. 2). The perovskite film cannot be treated as an intrinsic layer. As shown in Fig. 1, the simulation deviates evidently from the experimental results. So, the p-i-n junction is not suitable for the perovskite solar cells.

Our model is not a traditional p-i-n junction since we have explicitly included mobile ionic defects. We have assumed that these defects are non-ionising such that the intrinsic nature of the perovskite layer is preserved despite possibility for inhomogeneous ionic charge profiles. There are good grounds for this assumption, the abundance of evidence that ionic migration exists in these perovskites and calculations to suggest that the frontier energy levels of these ionic defects are beyond the band edges of MAPI (cited in the manuscript). Although the perovskite layer may be weakly n- or p- type (there are reports of both) the consensus in the literature is that the layer can be considered as intrinsic for the purpose of device models where the n- and p- type contact materials have carrier concentrations in great excess relative to the perovskite. (Richardson et al. EES, 10.1039/C5EE02740C; van Reenen et al. 10.1021/acs.jpcclett.5b01645)

We are not sure why the reviewer refers to Figure 2 here, as this figure is simply a scheme of the experimental timelines for the two optoelectronic transient measurements we have developed.

Our simulated JV curves in figure 1 clearly reproduce the key differences in hysteresis. The differences in V_{oc} between the measured and simulated cases arise because we have used a single set of simulation parameters for both device types despite the two architectures having differing contact energy levels (the values of which are uncertain but will influence the V_{oc} achievable; some variation in the band-to-band recombination rate constants is also likely between the architectures). Additionally we did not include features such as series resistance in the model, since this would not change the analysis conceptually, but would add unnecessary complexity. We felt that a comparison of simulations where only a single feature was changed (the recombination in the interfacial regions) gives a much stronger demonstration of the key ideas than a comparison of simulations where several parameters are varied in order to reproduce the V_{oc} by making arbitrary changes. We believe that our reproduction of the complex hysteresis effects by changing this single physical process in the model is one of the major strengths of the work.

Fortuitously, we also note that the new example of the top cathode device data we have used, shows a higher V_{oc} closer to that simulated, given the uncertainty in the device built-in potential this could be coincidental.

REVIEWERS' COMMENTS:

Reviewer #1 (Remarks to the Author):

The authors have made major improvements in the paper that answer my criticisms. It is a detailed and interesting study of perovskite cell characteristics that has the impact required for Nature Communications. With these improvements the paper can be accepted for publication subject to the mandatory corrections listed below.

1. The doping density assumed for the contact layers must be stated.
2. Lines 60-63 need rewriting. These statements contradict inclusion of a non-ionising mobile ionic species in their model and the the discussions of the influence of this species in lines 257, 274, 285, 299-356, 374-384, 389, 404-430.
3. Lines 105-107: the authors should suggest why a series resistance accounts for the discrepancy in fill factor.

Figure S3: the authors should check for consistency of line colours with figures in the main text and define the colours in the figure caption.

Reviewer #2 (Remarks to the Author):

Now I can confirm that this paper deserves publication in Nature Communication.

Reviewer #3 (Remarks to the Author):

The authors responded to my concerns and revised the manuscript carefully. The response to the simulation model of p-i-n is not so convinced, since the perovskite is a semiconducting layer and also a mixed ionic/electronic conductor. The revised manuscript probably can be published in Nature Communications.

REVIEWERS' COMMENTS:

Reviewer #1 (Remarks to the Author):

The authors have made major improvements in the paper that answer my criticisms. It is a detailed and interesting study of perovskite cell characteristics that has the impact required for Nature Communications. With these improvements the paper can be accepted for publication subject to the mandatory corrections listed below.

1. The doping density assumed for the contact layers must be stated.

Although this was stated in the supporting information, it is now stated clearly in the methods and where relevant in the text and figure captions.

2. Lines 60-63 need rewriting. These statements contradict inclusion of a non-ionising mobile ionic species in their model and the the discussions of the influence of this species in lines 257, 274, 285, 299-356, 374-384, 389, 404-430.

We believe that the reviewer misinterpreted our intended meaning for this sentence. We intended to convey that the available evidence in the literature would suggest two contradictory outlooks, and that there was evidence which contradicted the view that ion migration is responsible for hysteresis. Our data shows that this is not in fact the case. We have reworded the sentence to remove this ambiguity. The relevant passage now states:

“To a first approximation, ionic defect concentration and mobility in the bulk of the perovskite phase are not expected to be strongly influenced by the contacts, although the possibility that PCBM blocks ionic migration at grain boundaries has been proposed.³⁰ Superficially, these observations appear to undermine the viability of the ionic diffusion model for explaining hysteresis because the effect seems to be controlled by the contact material. However, recent simulations suggest that *J-V* hysteresis could only be reproduced if both ionic migration and recombination via interfacial traps were present in devices.¹⁶”

3. Lines 105-107: the authors should suggest why a series resistance accounts for the discrepancy in fill factor.

We have now discussed this in the main text.

“The effects of series resistance were not included, this accounts for the discrepancies in fill factor between experiment and simulation because the maximum power in the reverse scan would show a greater relative reduction than the forward scan due higher current densities.”

Figure S3: the authors should check for consistency of line colours with figures in the main text and define the colours in the figure caption.

We have modified the figures for consistency.